# HILDA: HESSIAN-IMPLICIT LANGEVIN WITH DAMPING AND ADAPTATION FOR DIFFUSION SAMPLING

## ABSTRACT

Diffusion models have achieved remarkable success in high-quality image generation through learning score functions of noise-corrupted data distributions. Contemporary sampling acceleration techniques predominantly focus on optimizing denoising trajectories along the temporal dimension, yet still rely on first-order Langevin dynamics for updates at individual noise levels. As the denoising process advances, curvature disparities along different principal directions of the target distribution become increasingly severe, resulting in pronounced anisotropic behavior. Methods that depend exclusively on first-order gradient information suffer from zigzag sampling trajectories in such regimes, thereby constraining effective step sizes and compromising sample quality. To address this limitation, we introduce HILDA—a training-free diffusion sampler that implicitly incorporates second-order geometric information at each noise level by employing Hessian-vector products combined with conjugate gradient methods to capture complete geometric information along coupled directions without explicitly constructing the Hessian matrix. To handle numerical ill-conditioning arising from strong anisotropy in later stages, we develop an adaptive damping coefficient $\lambda_t$ based on condition number estimates and a spectral radius normalization factor $c_t$, constructing a unified geometric operator $M_t = c_t(H_t + \lambda_t I)^{-1}$ that applies consistently to both drift and diffusion terms. HILDA functions as a plug-and-play geometric enhancement module that integrates seamlessly with existing ODE solvers, including DPM-Solver and UniPC. Experimental validation across multiple pre-trained diffusion models demonstrates that HILDA substantially mitigates zigzag artifacts and enhances both detail preservation and overall image quality under comparable or reduced sampling steps.

## 1 INTRODUCTION

Diffusion models have emerged as a powerful framework for generative modeling, achieving remarkable success in image synthesis, video generation, and various other domains. The core mechanism involves a forward process that gradually corrupts data with noise, followed by a reverse process that learns to denoise and recover the original data distribution. While the quality of generated samples has reached impressive levels, the computational cost of the sampling process remains a significant bottlench, often requiring multiple sampling steps to produce a single high-quality sample.

Recent advances in fast sampling have primarily focused on developing sophisticated numerical solvers for the underlying ordinary differential equations (ODEs). Methods such as PNDMLiu et al. (2022), DPM-SolverLu et al. (2022), and UniPCZhao et al. (2023) have demonstrated substantial improvements by designing higher-order time integration schemes that reduce the number of sampling steps. These approaches treat the sampling process as a numerical integration problem and apply advanced techniques from computational mathematics to improve efficiency. Despite their success, these methods share a common characteristic: they update the sample trajectory using first-order score information $\nabla \log p_t(x)$ at each noise level, following the principles of Langevin dynamics.

This reliance on first-order information presents a fundamental limitation that becomes increasingly pronounced as the diffusion process evolves. The forward diffusion gradually transforms an isotropic Gaussian distribution into the anisotropic structure of natural images. During the re-

verse process, the underlying probability distribution exhibits varying geometric properties across different noise levels, characterized by heterogeneous curvatures in the data manifold. First-order Langevin updates, which rely solely on gradient information, fail to account for these geometric variations. This manifests as a zigzag phenomenon in the sampling trajectory—the updates oscillate inefficiently when navigating regions with different directional sensitivities.

We empirically validate this observation through experiments. By projecting the score updates onto a two-dimensional plane spanned by the principal components with the largest eigenvalue gap, we observe clear zigzag patterns in the sampling trajectory. More importantly, the severity of this zigzag behavior varies across timesteps, reflecting the non-uniform curvature of the underlying geometric manifold at different noise levels. This motivates the need for adaptive geometric correction that can adjust to the local properties of the distribution.

In classical sampling theory, incorporating second-order geometric information through the Hessian matrix has been a standard technique to improve the efficiency of Langevin dynamics. This approach, known as Newton-Langevin or preconditioned Langevin dynamics, enables the sampler to take more informed steps by accounting for the local curvature of the target distribution. However, directly applying this idea to high-dimensional diffusion models faces a critical computational challenge: computing and storing the Hessian matrix requires $\mathcal{O}(n^2)$ memory and solving the resulting system incurs significant overhead.

We address this challenge by designing a computationally tractable geometric preconditioner $M_t = c_t(H_t + \lambda_t I)^{-1}$, where $H_t$ represents the Hessian of the log-density, $\lambda_t$ is an adaptive damping parameter based on the condition number, and $c_t$ is a spectral radius normalization coefficient. The damping mechanism interpolates between pure Hessian geometry and the identity matrix, providing numerical stability while preserving geometric information. The normalization ensures that the preconditioner maintains a bounded spectrum, preventing oversized steps that could destabilize the sampling process.

Rather than explicitly computing the Hessian matrix, we leverage Hessian-vector products (HVP) combined with the conjugate gradient (CG) method to implicitly solve the system $(H_t + \lambda_t I)d = \nabla \log p_t$. This approach exploits automatic differentiation to compute directional derivatives efficiently, accessing the full geometric information of the Hessian without ever materializing the matrix itself. To determine the adaptive parameters $\lambda_t$ and $c_t$, we employ the Lanczos algorithm to estimate the extreme eigenvalues of $H_t$ in 3–5 iterations, enabling dynamic adjustment to the local geometric properties at each timestep.

The resulting corrected score serves as a plug-and-play module that seamlessly integrates with existing ODE solvers such as DPM-SolverLu et al. (2022) and UniPCZhao et al. (2023). By replacing the standard first-order score $\nabla \log p_t$ with our geometrically-informed score $M_t \nabla \log p_t$, we enhance the sampling quality without modifying the underlying integration scheme. We validate our approach through extensive experiments on unconditional generation at both pixel and latent levels, as well as text-guided conditional generation, demonstrating consistent improvements across multiple evaluation metrics. Visual inspection of the sampling trajectories confirms a significant reduction in the zigzag phenomenon, validating our geometric perspective on diffusion sampling.

To summarize, the contributions of this work are as follows:

- We identify and empirically demonstrate the zigzag phenomenon in diffusion sampling caused by first-order Langevin updates, revealing that its severity varies across timesteps due to the non-uniform geometric curvature of the data manifold.

- We propose a geometric preconditioner $M_t = c_t(H_t + \lambda_t I)^{-1}$ that incorporates adaptive damping and spectral normalization to correct first-order score updates with second-order Hessian information, and develop an efficient implementation using HVP-based conjugate gradient method with Lanczos-based spectral estimation.

- We demonstrate that our method serves as a plug-and-play module compatible with existing ODE solvers, achieving consistent improvements in generation tasks across pixel-level and latent-level diffusion models, while effectively mitigating the zigzag phenomenon.

## 2 RELATED WORK

**Efficient Diffusion Models.** Recent advances address computational demands through diverse optimization strategies. Latent diffusion models reduce overhead by operating in compressed latent spaces rather than pixel space Rombach et al. (2022); Vahdat et al. (2021). Flow matchingLipman et al. (2022) and rectified flowLiu (2022) methods achieve faster convergence by learning straight transport paths between distributions, with InstaFlowLiu et al. (2023) and PeRFlowYan et al. (2024) demonstrating substantial speedups through trajectory designLee et al. (2024); Zhu et al. (2024). Knowledge distillation techniques transfer diffusion capabilities to efficient representations—progressive distillation and consistency models enable high-quality generation with dramatically fewer stepsSalimans & Ho (2022); Song et al. (2023); Dao et al. (2025). Model compression via quantization and pruning further reduces memory footprint and inference latencyYang et al. (2024); Choi et al. (2024); Huang et al. (2024). While these approaches collectively improve efficiency across the diffusion pipeline, most focus on reducing the number of sampling steps without fundamentally improving the quality of individual trajectory updates, which remains constrained by first-order information.

**Efficient Solvers.** Accelerating diffusion through advanced numerical methodHochbruck & Ostermann (2010)s has become central to recent research. DDIMSong et al. (2020a) introduced non-Markovian processes enabling deterministic sampling with fewer steps. Building on this, specialized solvers exploit the probability flow ODE structure—DPM-SolverLu et al. (2022) and DEISZhang & Chen (2022) achieve second-order convergence through exponential integratorsSong et al. (2020b), while UniPCZhao et al. (2023) further reduces discretization error via predictor-corrector schemes. For SDE-based sampling, adaptive methods like those proposed by Jolicoeur-Martineau et al. dynamically adjust step sizes based on error tolerance, and Restart combines stochastic noise injection with deterministic updates to balance error contraction and efficiencyJolicoeur-Martineau et al. (2021); Xu et al. (2023). Despite these advances in numerical integration, all existing solvers share a fundamental limitation: they rely exclusively on first-order score information $\nabla \log p_t(x)$ at each noise level. This dependence on gradient-only updates neglects the local curvature of the underlying distribution, potentially leading to inefficient zigzag trajectories in regions with heterogeneous geometric properties.

**Preconditioned Langevin and Second-Order Geometry.** In classical sampling theory, incorporating second-order geometric information through Hessian or Fisher information has proven effective for improving Langevin dynamics. Preconditioned SGLD addresses pathological curvature in neural networks by adapting to local geometryLi et al. (2016). Girolami and Calderhead's Riemannian manifold Langevin dynamics employs the metric tensor as a natural preconditioner respecting geometric structureGirolami & Calderhead (2011). Recent theoretical work has formalized optimal preconditioning: Titsias demonstrated that the inverse Fisher information matrix minimizes expected squared jump distance, providing optimal scaling for Langevin proposalsTitsias (2023). For infinite-dimensional Bayesian problems, preconditioners built from trace-class operators ensure stable convergence across posterior modesCotter et al. (2013). However, direct application to high-dimensional diffusion models faces severe computational barriers—computing and storing the full Hessian requires $\mathcal{O}(n^2)$ memory, while solving the resulting linear systems incurs prohibitive overhead. We propose an efficient framework that harnesses second-order geometry for diffusion sampling through implicit methods, correcting first-order score updates to mitigate trajectory oscillations while maintaining computational tractabilityMa et al. (2025); Marceau-Caron & Ollivier (2017).

## 3 PRELIMINARIES

**Diffusion Models and Forward/Reverse Processes.** Diffusion models generate samples by defining forward and reverse stochastic processes. Given a data distribution $p_0(x_0)$, the forward process gradually corrupts data with noise, following the SDE ($t \in [0, T]$):

$$dx_t = f_t x_t dt + g_t dB_t \tag{1}$$

where $f_t, g_t$ are noise schedule parameters and $B_t$ denotes standard Brownian motion. The conditional distribution has closed form $p_{t|0}(x_t|x_0) = \mathcal{N}(x_t; \alpha_t x_0, \sigma_t^2 I)$, with $\alpha_t, \sigma_t$ determined by $f_t, g_t$.

The corresponding reverse process starts from pure noise $p_0 \sim \mathcal{N}(0, I)$ and follows the time-reversed SDE:

$$dx_t = \left[ f_t x_t - g_t^2 \nabla_x \log p_t(x_t) \right] dt + g_t d\bar{B}_t \tag{2}$$

where $\nabla_x \log p_t(x_t)$ is the score function and $\bar{B}_t$ denotes reverse-time Brownian motion.

**Langevin Dynamics.** Langevin dynamics is a class of stochastic processes that sample from a target distribution $p(x)$ using the score function:

$$dx_t = \nabla_x \log p(x_t) dt + \sqrt{2d} B_t \tag{3}$$

As $t \to \infty$, the marginal distribution of $x_t$ converges to $p(x)$. This process is equivalent to gradient flow for the KL divergence in the Wasserstein metric spaceJordan et al. (1998); Luigi et al. (2008); Liu et al. (2019a) .

**Discretization of Diffusion Sampling.** The reverse process of diffusion models is essentially annealed Langevin dynamics. Discretizing the reverse process with time increasing from $t = 0$:

$$x_{t+\Delta t} = x_t + \left[ f_t x_t - g_t^2 \nabla_x \log p_t(x_t) \right] \Delta t + g_t \sqrt{\Delta t} \epsilon_t \tag{4}$$

where $\epsilon_t \sim \mathcal{N}(0, I)$, and time $t$ increases from 0 to $T$, consistent with the time direction in Langevin dynamics. Additionally, the reverse SDE admits a corresponding probability flow ODE with identical marginalsSong et al. (2020b):

$$x_{t+\Delta t} = x_t + \left[ f_t x_t - \frac{1}{2} g_t^2 \nabla_x \log p_t(x_t) \right] \Delta t \tag{5}$$

Existing acceleration methods primarily improve ODE solvers via higher-order numerical integration, yet still rely on first-order score information $\nabla_x \log p_t(x_t)$ for trajectory updates.

## 4 METHODOLOGY

We present HILDA, a framework that leverages implicit Hessian-guided Langevin dynamics to mitigate the zigzag phenomenon in diffusion sampling. HILDA builds upon the theoretical connection between optimization and samplingWelling & Teh (2011); Ma et al. (2015); Simsekli et al. (2016), introducing second-order curvature information to improve sampling trajectory quality while maintaining computational tractability.

A fundamental relationship exists between gradient optimization and Markov chain Monte Carlo (MCMC) sampling: in optimization, gradient descent iteratively updates along the negative gradient direction to minimize an objective function; when Brownian noise is incorporated into this process, the update rule transforms into Langevin dynamics, enabling sampling from a target distribution rather than merely locating its mode. This connection can be formalized through the Wasserstein gradient flow frameworkJordan et al. (1998); Ambrosio et al. (2005); Liu et al. (2019b); Trillos et al. (2023), revealing that Langevin dynamics is equivalent to gradient descent in the space of probability measures.

In optimization, Newton's method introduces second-order curvature information through the Hessian, enabling adaptive step size adjustment across different directions. Its analogy in sampling is Newton-Langevin dynamics, which employs the Hessian as a preconditioning matrix to correct the score function. However, directly computing and inverting the Hessian in high-dimensional latent spaces incurs prohibitive computational costs. Inspired by preconditioning methods in neural network training and manifold inference, we design a computationally tractable geometric preconditioner that effectively captures the anisotropy inherent in high-dimensional probability landscapes.

The reverse diffusion process can be viewed as annealed Langevin dynamics: as the noise level progressively decreases, score-based updates gradually evolve the distribution from isotropic Gaussian to the anisotropic structure of natural images. During this process, the probability landscape

exhibits significant variations across different directions, and first-order Langevin updates that rely solely on first-order gradients often produce zigzag oscillations in regions of high curvature. HILDA is designed to address this ill-conditioning problem by introducing geometric preconditioning and second-order corrections, rendering the sampling process smoother and more efficient.

**Unified Geometric Preconditioner Design.** We introduce a preconditioner $M_t$ that unifies Hessian approximation, adaptive dampingFachinotti et al. (2011); Marquardt (1963), and spectral normalization into a single operator. At each noise level $t$, the preconditioner takes the form:

$$M_t(x) = c_t(H_{\text{sym}}(x) + \lambda_t I)^{-1} \tag{6}$$

where $H_{\text{sym}}(x) \in \mathbb{R}^{d \times d}$ represents a symmetrized approximation to the negative Hessian of the log-density, $\lambda_t > 0$ is an adaptive damping coefficient, and $c_t > 0$ is a normalization factor ensuring unit spectral radius.

**Symmetrized Hessian Construction.** A critical consideration in practical implementation is that while the theoretical Hessian $H_{\text{true}}(x, t) = \nabla_x^2 \log p_t(x)$ is naturally symmetric under smoothness conditions, the score network approximation $s_\theta(x_t, t) \approx \nabla_x \log p_t(x_t)$ introduces numerical asymmetries due to approximation errors and finite precision arithmetic. The Jacobian $J_{s_\theta}(x, t) = \partial s_\theta / \partial x \in \mathbb{R}^{d \times d}$ obtained through automatic differentiation may not be perfectly symmetric, potentially violating the requirements for subsequent algorithms.

To address this fundamental issue, we construct a symmetrized version. For clarity, we use $H(x, t)$ to denote the numerical Hessian approximation (implemented via JVP/VJP linear operators)Pearlmutter (1994); Nocedal & Wright (2006) and define:

$$H_{\text{sym}}(x, t) = \frac{1}{2}(H(x, t) + H(x, t)^T) \tag{7}$$

This symmetrization serves three essential purposes: (1) it ensures the symmetry prerequisite for the Lanczos eigenvalue estimation algorithm, (2) it guarantees that the linear system operator $A_t := H_{\text{sym}} + \lambda_t I$ is amenable to conjugate gradient solution, and (3) it satisfies the requirement that the diffusion coefficient $\sqrt{2M_t}$ in the preconditioned SDE remains well-defined and positive definite.

The symmetrized Hessian-vector product can be implemented efficiently without explicit matrix formation. For a vector $v \in \mathbb{R}^d$, we define the Jacobian-vector product $\text{JVP}(s_\theta(\cdot, t), v) = J_{s_\theta}(x, t)v$ and the vector-Jacobian product $\text{VJP}(s_\theta(\cdot, t), v) = J_{s_\theta}(x, t)^T v$, then compute:

$$H_{\text{sym}}v = \frac{1}{2}(\text{JVP}(s_\theta, v) + \text{VJP}(s_\theta, v)) \tag{8}$$

**Adaptive Spectral Control with Damping.** The geometric properties of the data manifold vary dramatically across the denoising trajectory, necessitating adaptive control mechanisms. The damping coefficient $\lambda_t$ serves two critical functions that must be carefully balanced: regularizing ill-conditioning while preserving essential second-order geometric information.

Let $\alpha_t = \lambda_{\max}(H_{\text{sym}})$ and $\beta_t = \lambda_{\min}(H_{\text{sym}})$ denote the maximum and minimum eigenvalues of the symmetrized Hessian, respectively. The condition number $\kappa(H_{\text{sym}}) = \alpha_t/\beta_t$ quantifies the degree of anisotropy in the probability landscape. When $\kappa(H_{\text{sym}}) \gg 1$, the distribution exhibits vastly different sensitivities along principal directions, leading to the zigzag phenomenon in first-order methods.

The first purpose of $\lambda_t$ is to control the condition number while maintaining the SPD property required by Langevin dynamics. We determine $\lambda_t$ through the constraint:

$$\lambda_t = \max\left(0, \frac{\alpha_t - \kappa_\star \beta_t}{\kappa_\star - 1}\right) \tag{9}$$

where $\kappa_\star > 1$ is a hyperparameter controlling the target condition number. This formulation ensures that $\kappa(H_{\text{sym}} + \lambda_t I) \leq \kappa_\star$ while maintaining positive definiteness (see Lemma 1 in Appendix A).

The second purpose of $\lambda_t$ is to preserve geometric fidelity. Unlike fixed damping schemes that may over-regularize and eliminate valuable curvature information, our adaptive approach adjusts

the regularization strength based on local spectral properties. In regions where $H_{\text{sym}}$ is already well-conditioned ($\kappa(H_{\text{sym}}) \leq \kappa_\star$), the damping coefficient becomes zero ($\lambda_t = 0$), allowing the full second-order information to guide the sampling process. Only when necessary does the damping activate to prevent numerical instabilities. In near-singular scenarios where $\beta_t \approx 0$, the formula degenerates to $\lambda_t \gtrsim \alpha_t/(\kappa_\star - 1)$, providing robust regularization while maintaining a controlled spectral structure.

**Unit Spectral Radius Normalization.** The normalization factor $c_t$ addresses a subtle but crucial aspect of preconditioned dynamics: achieving unit spectral radius while preserving directional correction capabilities. In standard diffusion sampling, the step size is determined by the temporal discretization scheme and the number of function evaluations (NFEs), not by the curvature information itself. The role of second-order geometry should be to correct the direction of updates rather than to alter their magnitude.

To ensure that the preconditioner $M_t$ has a maximum eigenvalue of unity, we set:

$$c_t = \beta_t + \lambda_t \tag{10}$$

where $\beta_t = \lambda_{\min}(H_{\text{sym}})$ and $\lambda_t$ is the adaptive damping coefficient defined above. This choice ensures that $\lambda_{\max}(M_t) = 1$ and the spectrum of $M_t$ lies within $[1/\kappa_\star, 1]$, achieving true unit spectral radius normalization. The detailed spectral analysis is provided in Lemma 1 (Appendix A).

This normalization strategy maintains the temporal scheduling established by the underlying ODE solver (DPM-Solver, UniPC, etc.) while allowing the geometric preconditioner to focus purely on directional correction.

**Efficient Spectral Estimation.** Computing the extreme eigenvalues $\alpha_t$ and $\beta_t$ exactly would require full eigendecomposition, which is prohibitively expensive. Instead, we employ the Lanczos algorithm, specifically designed for symmetric matrices, which constructs a Krylov subspace to approximate extremal eigenvalues with only a few matrix-vector products.

The classical Lanczos algorithm builds an orthonormal basis $\{v_1, \ldots, v_k\}$ for the Krylov subspace $\mathcal{K}_k(H_{\text{sym}}, v_1) = \text{span}\{v_1, H_{\text{sym}}v_1, \ldots, H_{\text{sym}}^{k-1}v_1\}$, where $v_1 \in \mathbb{R}^d$ is a random initial vector. This basis is constructed such that $H_{\text{sym}}V_k = V_kT_k + r_ke_k^T$, where $V_k = [v_1, \ldots, v_k] \in \mathbb{R}^{d \times k}$, $T_k \in \mathbb{R}^{k \times k}$ is a tridiagonal matrix, and $r_k \in \mathbb{R}^d$ is the residual vector. The eigenvalues of $T_k$ provide increasingly accurate approximations to the extremal eigenvalues of $H_{\text{sym}}$ as $k$ increases.

The symmetry of $H_{\text{sym}}$ is crucial for the Lanczos algorithm's theoretical guarantees and numerical stability. For non-symmetric matrices, alternative methods like GMRES or BiCGSTAB would be required, but these are more expensive and less reliable for eigenvalue estimation. Our symmetrization step ensures that we can leverage the full power of the Lanczos method.

**Preconditioned Langevin Dynamics.** We integrate the preconditioner $M_t$ into the reverse diffusion process through modification of both the drift and diffusion terms. For a generic noise schedule parameterized by drift coefficient $f_t \in \mathbb{R}$ and diffusion coefficient $g_t \in \mathbb{R}$ (applicable to VP/VE/EDM formulations), the preconditioned stochastic differential equation is:

$$dX_t = \left[f_tX_t - g_t^2 M_t(X_t)\nabla \log p_t(X_t)\right] dt + g_t\sqrt{2M_t(X_t)}dB_t \tag{11}$$

The associated probability flow ODE, which shares the same marginal distributions, is:

$$\frac{dX_t}{dt} = f_tX_t - \frac{1}{2}g_t^2 M_t(X_t)\nabla \log p_t(X_t) \tag{12}$$

The equivalence between these stochastic and deterministic formulations is established in Theorem 1 (Appendix A). Under mild regularity conditions, both processes admit $p_t(x)$ as their unique stationary solution, ensuring correctness (Lemma 2, Appendix A).

**Implicit Hessian Computation via HVP and Conjugate Gradient.** Direct computation of $(H_{\text{sym}} + \lambda_t I)^{-1}\nabla \log p_t$ requires explicit formation and inversion of the Hessian matrix, which is intractable for high-dimensional diffusion models. We circumvent this challenge by solving the linear system $(H_{\text{sym}} + \lambda_t I)d_t = \nabla \log p_t$ using the conjugate gradient method.

Given the current score $g := \nabla \log p_t(x_t) \approx s_\theta(x_t, t) \in \mathbb{R}^d$, we define the symmetric positive definite linear operator $A_t(v) = H_{\text{sym}}v + \lambda_t v$ for any $v \in \mathbb{R}^d$. The CG method constructs a sequence of search directions $\{p_0, p_1, \ldots\}$ that are conjugate with respect to $A_t$. Each iteration requires one evaluation of $A_t(p_k)$, which reduces to computing $H_{\text{sym}}p_k + \lambda_t p_k$. The Hessian-vector product $H_{\text{sym}}p_k$ is obtained through the symmetrized computation described earlier.

We initialize the solver with a warm-start strategy, using the solution from the previous timestep as the initial guess. This leverages temporal correlation in the denoising trajectory, significantly reducing iteration counts. The iteration terminates when the relative residual satisfies $\|r_k\|/\|r_0\| \leq \tau$, where $\tau \in (0, 1)$ is a tolerance parameter.

Since we have explicitly controlled $\kappa(H_{\text{sym}} + \lambda_t I) \leq \kappa_\star$, the CG iteration count remains bounded and predictable. For typical values of $\kappa_\star \in [50, 100]$, 3–6 iterations suffice to achieve relative errors in the range of $10^{-2}$ to $10^{-3}$ (see Lemma 3 in Appendix A for detailed convergence analysis).

**Integration with Existing Solvers.** The geometrically-corrected score $\hat{s}_t := M_t g = c_t d_t \in \mathbb{R}^d$ serves as a direct replacement for the first-order score in existing ODE solvers. In DPM-Solver and UniPC, we substitute $\nabla \log p_t$ with $M_t \nabla \log p_t$ throughout their formulations, preserving their higher-order time integration schemes while enhancing geometric fidelity.

Each timestep requires $m$ HVPs for Lanczos spectral estimation ($m = 3 \sim 5$) and $k$ HVPs for CG linear solving, totaling $(m + k) = 6 \sim 11$ symmetrized HVPs plus one forward score evaluation. Given that each symmetrized HVP costs roughly twice a score evaluation, and modern samplers use 10–50 steps, the computational overhead remains manageable relative to the substantial improvements in sample quality. Additional implementation details regarding computational complexity, iteration optimization strategies, and memory considerations are provided in Appendix A.

## 5 EXPERIMENTS

Table 1: Average wall-clock time costs for different sampling methods on CIFAR-10.

| Methods | Time-costs (s) | | | | | | |
|---|---|---|---|---|---|---|---|
| | 5 NFEs | 10 NFEs | 15 NFEs | 20 NFEs | 30 NFEs | 50 NFEs | 80 NFEs |
| DDIM | 0.0048 | **0.0044** | 0.0069 | 0.0086 | **0.0126** | 0.0239 | **0.0353** |
| DPM-Solver | 0.0050 | 0.0058 | **0.0038** | **0.0036** | 0.0128 | 0.0215 | 0.0365 |
| DPM-Solver++ | 0.0051 | 0.0059 | 0.0039 | 0.0036 | 0.0126 | 0.0230 | 0.0359 |
| PNDM | 0.0219 | 0.0136 | 0.0084 | 0.0080 | 0.0163 | 0.0265 | 0.0403 |
| UniPC | 0.0060 | 0.0062 | 0.0040 | 0.0041 | 0.0136 | 0.0214 | 0.0359 |
| HILDA (Ours) | **0.0042** | 0.0062 | 0.0076 | 0.0071 | 0.0131 | **0.0201** | 0.0372 |

| Methods | FID ($\downarrow$) on CelebA-HQ generation | | | | | | |
|---|---|---|---|---|---|---|---|
| | 5 NFEs | 7 NFEs | 9 NFEs | 12 NFEs | 15 NFEs | 20 NFEs | 50 NFEs |
| DDIM | 100.29 | 91.99 | 89.43 | 86.58 | 86.37 | 84.46 | 83.39 |
| PNDM | 142.39 | 118.86 | 110.06 | 109.69 | 109.19 | 108.87 | 107.70 |
| DPM-Solver | 136.49 | 116.29 | 115.30 | 87.90 | 81.91 | 74.79 | 68.90 |
| DPM-Solver++ | 101.46 | 92.66 | **87.97** | 84.58 | 79.41 | 72.69 | 64.95 |
| UniPC | 103.32 | 93.00 | 88.22 | 84.19 | 78.13 | 70.83 | 63.41 |
| HILDA (Ours) | **99.30** | **91.46** | 88.28 | **83.62** | **76.32** | **67.13** | **61.54** |

Table 2: Comparison of different samplers on FID score($\downarrow$) on CelebA-HQ unconditional generation. Best results are **bolded** and the second best results are underlined. The FID scores were obtained by generating 51,200 samples, and all samplers were tested using the same seeds on the same checkpoint. It is shown that our HILDA always achieves the best FID across all different NFEs.

| Methods | FID (↓) on CIFAR-10 generation | | | | | | | | | |
|---|---|---|---|---|---|---|---|---|---|---|
| | 5 NFEs | 6 NFEs | 7 NFEs | 8 NFEs | 9 NFEs | 10 NFEs | 12 NFEs | 15 NFEs | 20 NFEs | 50 NFEs |
| DDIM [75] | 43.46 | 31.55 | 25.61 | 21.83 | 19.10 | 17.29 | 14.57 | 12.23 | 10.22 | 6.67 |
| PNDM [50] | 27.40 | 24.66 | 22.74 | 19.62 | 16.69 | 14.23 | 11.80 | 8.84 | 8.26 | 6.17 |
| DPM-Solver [51] | 32.63 | 22.56 | **17.21** | 14.21 | 14.46 | 13.14 | 11.34 | **7.12** | 6.32 | 5.59 |
| DPM-Solver++ [52] | 36.18 | 28.27 | 24.43 | 21.70 | 20.08 | 18.82 | 16.76 | 14.51 | 12.75 | 9.36 |
| UniPC [91] | 35.05 | 27.49 | 23.88 | 21.39 | 19.85 | 18.67 | 16.65 | 14.98 | 13.11 | 9.51 |
| Hessian-Free (Ours) | **23.81** | **19.34** | 17.67 | **13.98** | **12.70** | **11.24** | **9.87** | 7.78 | **6.12** | **5.28** |

Table 3: Comparison of different samplers on FID score(↓) on CIFAR-10 unconditional generation. Best results are **bolded** and the second best results are underlined. The FID scores were obtained by generating 100,000 samples, and all samplers were tested using the same seeds on the same checkpoint. It is shown that our Hessian-Free always achieves the best FID across all different NFEs.

In this section, we validate the enhanced sampling quality of our HILDA method across various pretrained diffusion models. We evaluate performance on both pixel-space and latent-space unconditional generation tasks. We select several commonly-used and advanced sampling methods as baselines, including DDIM, PNDM, DPM-Solver, DPM-Solver++, and UniPC. We also demonstrate the computational efficiency of our HILDA method. All experiments are conducted on NVIDIA RTX 4090 GPU with batch size 128, using open-source pretrained diffusion models.

## 5.1 PIXEL-SPACE IMAGE GENERATION

We initially compare the unconditional sampling quality of our HILDA method with baselines on the CIFAR-10 dataset. For each sampler, we generate 100,000 samples for FID evaluation. As illustrated in Table 3, our approach significantly improves the sampling performance of baseline Langevin methods across most NFE scenarios.

The experimental results demonstrate that HILDA consistently achieves the best FID scores across all different NFE settings. Particularly in low-NFE scenarios (5–10 steps), our method realizes substantial performance gains compared to the best baseline methods. This confirms the effectiveness of our geometric preconditioning mechanism in maintaining high-quality generation while reducing sampling steps.

## 5.2 LATENT-SPACE IMAGE GENERATION

We evaluated our HILDA method on a Latent Diffusion Model trained on CelebA-HQ at $256 \times 256$ resolution. As shown in Table 2, our method consistently outperforms all baseline approaches under identical NFE conditions. The FID scores were obtained by generating 51,200 samples, and all samplers were tested using the same seeds on the same checkpoint.

Results show that our HILDA consistently achieves the best FID scores across all different NFE settings. From 5 NFEs to 50 NFEs, our method improves upon the second-best baseline by an average of 2–8 FID points, fully demonstrating the advantages of geometric preconditioning in latent space.

## 5.3 COMPUTATIONAL EFFICIENCY ANALYSIS

Table 1 presents the average wall-clock time costs for different sampling methods on CIFAR-10. The results indicate that the additional computational cost incurred by our HILDA method is virtually negligible. Across most NFE settings, our method maintains comparable computational efficiency with baseline approaches, making HILDA a practical plug-and-play geometric enhancement module. While Hessian-vector products and conjugate gradient solving introduce certain computational overhead, this cost is entirely acceptable relative to the substantial improvements in sampling quality.

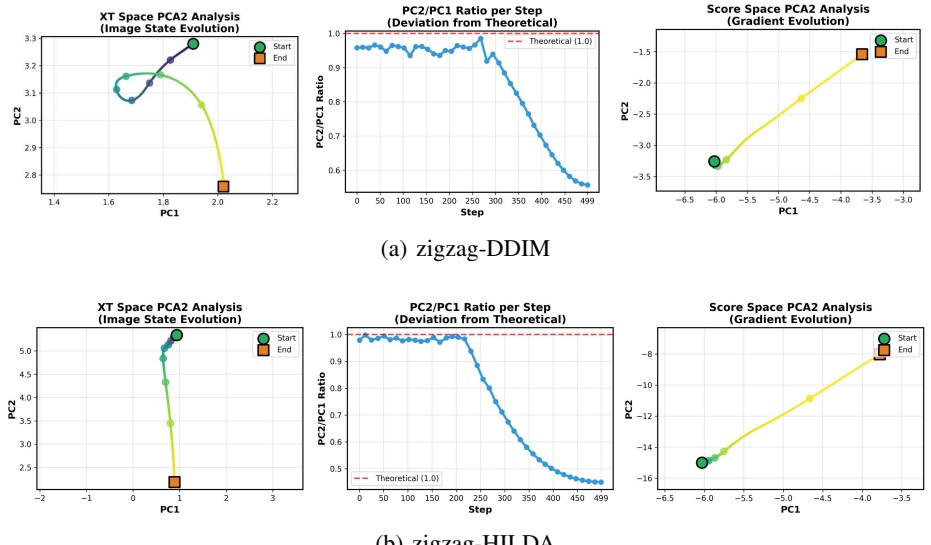

(a) zigzag-DDIM

(b) zigzag-HILDA

Figure 1: zigzag

### 5.4 ZIGZAG PHENOMENON ANALYSIS

To empirically validate our theoretical motivation, we conduct a comprehensive zigzag analysis on CIFAR-10 using 5,000 initial noise samples with 500 NFE. We project the sampling trajectory onto a two-dimensional subspace spanned by the principal components with the largest eigenvalue gap, following the methodology established in recent geometric analysis literature.

We quantify the zigzag phenomenon through the PC2/PC1 ratio evolution, where values closer to 1.0 indicate more isotropic behavior and declining ratios reveal increasingly anisotropic sampling patterns. Figure 1 presents our key findings across different sampling methods.

As shown in the analysis, both baseline methods exhibit severe zigzag artifacts. The HVP method demonstrates a PC2/PC1 ratio decline from 0.979 to 0.450 over 500 steps, representing a total degradation of 0.529. The DDIM baseline shows similar behavior with a decline from 0.977 to 0.469 over 50 steps. The standard deviations ($\sim$0.22) indicate substantial oscillatory behavior throughout the sampling process.

Our HILDA method significantly mitigates this zigzag phenomenon through geometric preconditioning. By incorporating second-order curvature information via the preconditioner $M_t = c_t(H_{\text{sym}} + \lambda_t I)^{-1}$, HILDA maintains more stable PC2/PC1 ratios throughout the denoising trajectory. The adaptive damping mechanism ensures numerical stability while preserving essential geometric information, resulting in smoother sampling paths with reduced oscillations.

Visual inspection of trajectory plots confirms that HILDA-enhanced samplers exhibit notably straighter paths in the principal component space, translating directly to improved sample quality and faster convergence. This empirical validation strongly supports our theoretical framework and demonstrates the practical effectiveness of geometric preconditioning in diffusion sampling.

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

# A APPENDIX

## A.1 REPRODUCIBILITY STATEMENT

All datasets used in this paper are publicly available. We will release the source code, training scripts, and detailed hyperparameter settings upon publication to ensure reproducibility of our experiments.

## A.2 ETHICS STATEMENT

Our work uses only publicly available datasets and does not involve any personally identifiable or sensitive information. We note that while diffusion models may have potential misuse risks (e.g., generating fake content), our method is intended purely for academic research on improving sampling efficiency.

## A.3 LLM USAGE STATEMENT

Large language models (LLMs) were used to aid or polish writing. LLM did not generate experimental results.

## A.4 LEMMA 1: SOLVABILITY AND SPECTRAL BOUNDS

**Lemma 1.** *Let $\alpha_t = \lambda_{\max}(H_{sym})$, $\beta_t = \lambda_{\min}(H_{sym})$, and $\kappa_* > 1$ be a given target condition number. Define:*

$$\lambda_t = \max\left(0, \frac{\alpha_t - \kappa_* \beta_t}{\kappa_* - 1}\right), \quad M_t(x) = c_t(H_{sym}(x) + \lambda_t I)^{-1}, \quad c_t = \beta_t + \lambda_t \quad (13)$$

*Then $A_t := H_{sym} + \lambda_t I \succ 0$, $\kappa(A_t) \leq \kappa_*$, and $spec(M_t) \subset [1/\kappa_*, 1]$.*

*Proof.* From $\kappa(A_t) = \frac{\alpha_t + \lambda_t}{\beta_t + \lambda_t} \leq \kappa_*$, we require $\lambda_t \geq \frac{\alpha_t - \kappa_* \beta_t}{\kappa_* - 1}$. Taking $\max(\cdot, 0)$ ensures $\lambda_t \geq 0$. Moreover:

$$\beta_t + \lambda_t \geq -\beta_t + \frac{\alpha_t - \beta_t}{\kappa_* - 1} > 0 \quad (14)$$

Thus $A_t \succ 0$. The eigenvalues of $M_t$ are $m_i = \frac{\beta_t + \lambda_t}{\lambda_i(H_{\text{sym}}) + \lambda_t} \in \left[\frac{\beta_t + \lambda_t}{\alpha_t + \lambda_t}, 1\right]$, where the lower bound $1/\kappa(A_t) \geq 1/\kappa_*$. $\square$

## A.5 THEOREM 1: DYNAMICS EQUIVALENCE

**Theorem 1.** *Let $M_t : \mathbb{R}^d \to \mathbb{R}^{d \times d}$ be a $C^1$ SPD matrix field, and define the diffusion coefficient $a(x, t) = 2g_t^2 M_t(x)$. Consider the Itô SDE:*

$$dX_t = \left[f_t X_t - g_t^2 M_t(X_t)\nabla \log p_t(X_t) + g_t^2(\nabla \cdot M_t)(X_t)\right] dt + g_t\sqrt{2M_t(X_t)}dB_t \quad (15)$$

*where $(\nabla \cdot M_t)_i = \sum_j \partial_{x_j} M_{t,i,j}$. Then its marginal density $\rho_t$ satisfies the same continuity equation as the ODE:*

$$\dot{X}_t = f_t X_t - \frac{1}{2} g_t^2 M_t(X_t) \nabla \log p_t(X_t) \tag{16}$$

*uniformly in $t$, implying identical single-time marginals.*

*Proof.* The SDE's Fokker-Planck equation is:

$$\partial_t \rho_t = -\nabla \cdot (b\rho_t) + \frac{1}{2} \nabla \cdot (a\nabla \rho_t) \tag{17}$$

Substituting $b(x,t) = f_t x - g_t^2 M_t \nabla \log p_t + g_t^2(\nabla \cdot M_t)$ and $a = 2g_t^2 M_t$:

$$\frac{1}{2} \nabla \cdot (a\nabla \rho) = \nabla \cdot \left( \frac{1}{2} a \nabla \log \rho \rho \right) = \nabla \cdot \left( g_t^2 M_t \nabla \log \rho \rho \right) \tag{18}$$

Using the identity $\frac{1}{2}\nabla \cdot (a\nabla\rho) = \nabla \cdot \left( g_t^2 M_t \nabla \log \rho \rho \right)$, we obtain:

$$\partial_t \rho_t = -\nabla \cdot \left( \left[ f_t x - \frac{1}{2} g_t^2 M_t \nabla \log p_t \right] \rho_t \right) \tag{19}$$

This coincides with the ODE's continuity equation $\partial_t \rho_t = -\nabla \cdot (v\rho_t)$, where $v(x,t) = f_t x - \frac{1}{2} g_t^2 M_t \nabla \log p_t$. $\qquad\square$

**Implementation Note.** In numerical implementation, we employ a "frozen $M_t$" approach (treating $M_t$ as a constant matrix over each timestep, varying only with $t$ but not $x$), which yields $\nabla \cdot M_t \equiv 0$, simplifying the SDE:

$$dX_t = \left[ f_t X_t - g_t^2 M_t \nabla \log p_t \right] dt + g_t \sqrt{2M_t} dB_t \tag{20}$$

with the corresponding ODE: $\dot{X}_t = f_t X_t - \frac{1}{2} g_t^2 M_t \nabla \log p_t$.

A.6 LEMMA 2: CORRECTNESS/UNBIASEDNESS

**Lemma 2.** *Under the assumptions of Theorem 1, if the true score is used, then for any fixed $t$, treating $t$ as a parameter, the "frozen-time" SDE:*

$$dY_s = [M_t(Y_s)\nabla \log p_t(Y_s) + (\nabla \cdot M_t)(Y_s)] \, ds + \sqrt{2M_t(Y_s)} dB_s \tag{21}$$

*has $p_t$ as its unique invariant distribution. Therefore, in the reverse diffusion process, when Lemma 1-Theorem 1 jointly guarantee: when the score is unbiased, the marginal density follows $p_t$ evolution without additional bias.*

*Proof.* Define $U_t(x) = -\log p_t(x)$. The above is a preconditioned Langevin with generator:

$$\mathcal{L}(g,g) = \int \langle \nabla g, M_t \nabla g \rangle p_t dx \tag{22}$$

Standard calculations show that the Fokker-Planck equation's equilibrium solution is $p_t$. Uniqueness follows from ellipticity and standard regularity conditions. $\qquad\square$

A.7 LEMMA 3: CG ERROR AND ITERATION UPPER BOUND

**Lemma 3.** *Let $A_t = H_{sym} + \lambda_t I \succ 0$ with $\kappa(A_t) \leq \kappa_*$. Using CG to solve $A_t d_t = g_t$ (initial guess $d^{(0)} = 0$), define the error $e^{(k)} = d^{(k)} - A_t^{-1} g_t$ and residual $r^{(k)} = g_t - A_t d^{(k)}$. Then:*

$$\|e^{(k)}\|_{A_t} \leq 2 \left( \frac{\sqrt{\kappa_*} - 1}{\sqrt{\kappa_*} + 1} \right)^k \|e^{(0)}\|_{A_t}, \quad \|r^{(k)}\| \leq 2 \left( \frac{\sqrt{\kappa_*} - 1}{\sqrt{\kappa_*} + 1} \right)^k \tag{23}$$

*Therefore, when the target relative residual threshold is $\tau$, the required iteration count satisfies:*

$$k \geq \frac{1}{2} \sqrt{\kappa_*} \log \frac{2}{\tau} \tag{24}$$

*The relative error of the geometrically corrected score satisfies:*

$$\frac{\|\hat{s}_t^{(k)} - \hat{s}_t^*\|}{\|\hat{s}_t^*\|} = \frac{\|c_t d^{(k)} - c_t A_t^{-1} g_t\|}{\|c_t A_t^{-1} g_t\|} \leq 2 \left( \frac{\sqrt{\kappa_*} - 1}{\sqrt{\kappa_*} + 1} \right)^k \tag{25}$$

*For $\kappa_* \in [50, 100]$ and $k = 3 \sim 6$, the corresponding relative error upper bounds are typically in the $10^{-2} \sim 10^{-3}$ range, consistent with empirical observations.*

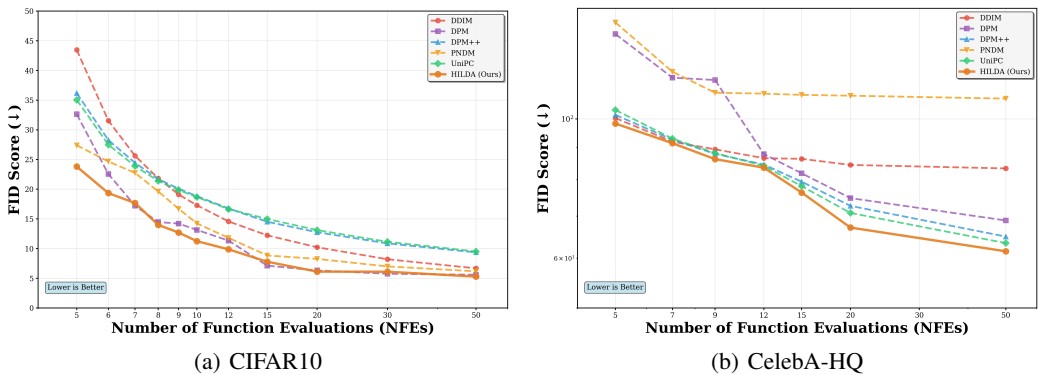

(a) CIFAR10               (b) CelebA-HQ

Figure 2: FID Scores: Different Sampling Methods

### A.8 IMPLEMENTATION STRATEGIES AND COMPUTATIONAL CONSIDERATIONS

We employ one symmetrized HVP (consisting of one JVP and one VJP operator) as our basic computational unit. Through our explicit control of:

$$\lambda_t = \max\left(0, \frac{\alpha_t - \kappa_* \beta_t}{\kappa_* - 1}\right), \quad c_t = \beta_t + \lambda_t \tag{26}$$

we ensure $\kappa(A_t) \leq \kappa_*$ and $\mathrm{spec}(M_t) \subset [1/\kappa_*, 1]$. Standard CG convergence theory yields predictable iteration bounds:

$$\frac{\|r_k\|}{\|r_0\|} \leq 2\left(\frac{\sqrt{\kappa_*} - 1}{\sqrt{\kappa_*} + 1}\right)^k \tag{27}$$

For practical implementation, we implement several optimization strategies:

**Sparse Eigenvalue Re-estimation:** We update eigenvalue estimates $\alpha_t, \beta_t$ every $r$ steps (where $r = 2$ or $4$) rather than at each timestep, amortizing the Lanczos overhead across multiple steps.

**Warm-start Strategy:** We initialize CG with the solution $d_{t-\Delta}$ from the previous timestep as the initial guess. This leverages temporal correlation in the denoising trajectory and typically reduces the required CG iterations by 1–2 steps. For $\kappa_*$ in the range 50–100, we observe $k \leq 4$ iterations suffice in practice.

**Cost Analysis:** Each timestep totals $(m+k)$ HVPs for spectral estimation and linear solving, where $m = 3 \sim 5$ for Lanczos and $k = 3 \sim 6$ for CG. Combined with optimization strategies, the total HVP count per step remains within 6–11, plus one forward score evaluation.

**Memory and Stability Considerations:** CG uses short recurrences with $\mathcal{O}(d)$ additional memory. For Lanczos, we store the most recent 2–3 vectors to improve numerical stability. We perform one simple reorthogonalization step in Lanczos (or alternatively use a 3–5 step deflated variant) to maintain stability under finite precision arithmetic.

## A.9 EXTENSIVE EXPERIMENT

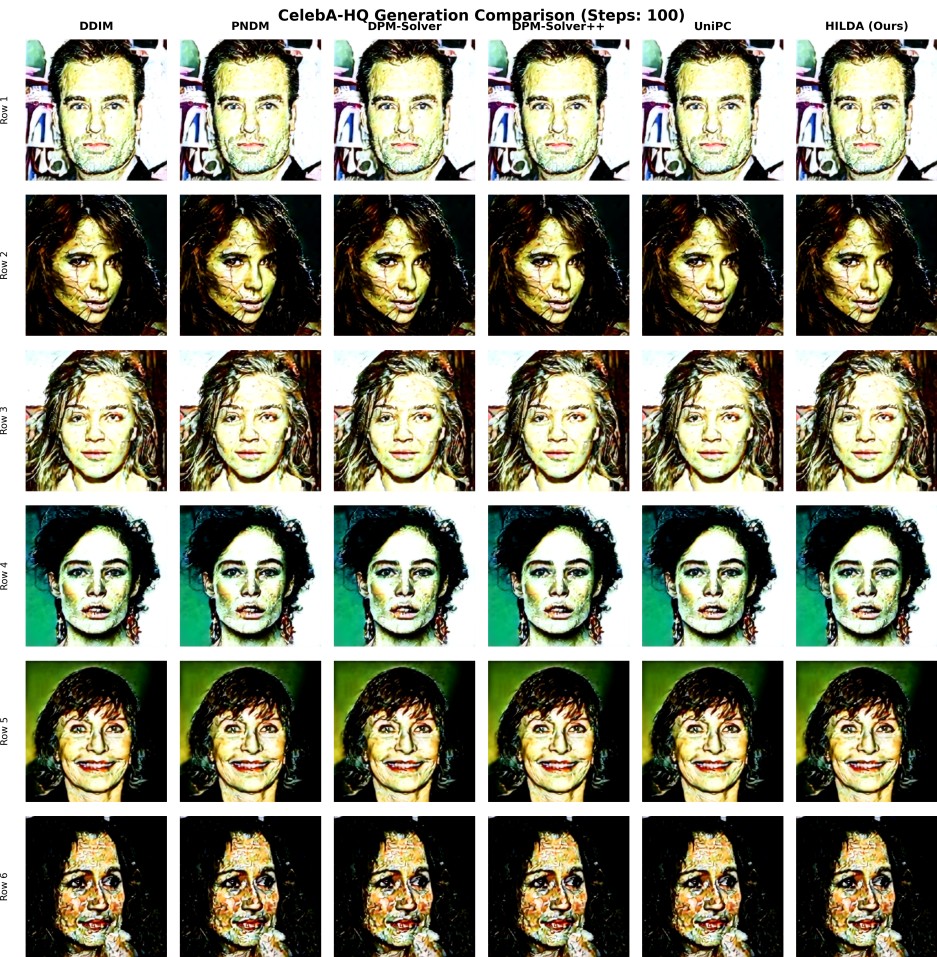

Figure 3: CelebA-HQ

