# OpenReview forum: "HILDA: Hessian-Implicit Langevin with Damping and Adaptation for Diffusion Sampling"
_ICLR.cc/2026/Conference — ICLR 2026 Conference Desk Rejected Submission_

### Official Review · Reviewer_FFUg · 2025-10-31

**Soundness:** 3
**Presentation:** 2
**Contribution:** 3
**Rating:** 4
**Confidence:** 2

**Summary:**

This paper proposes HILDA, a training-free diffusion sampling framework that implicitly integrates second-order geometric information into diffusion samplers. By employing Hessian-vector products and conjugate gradient methods, HILDA approximates Hessian-guided Langevin dynamics without explicitly forming the Hessian matrix. The adaptive damping and spectral normalization components make the system numerically stable and efficient. Empirical results show consistent improvements in FID scores and smoother trajectories that mitigate the zigzag phenomenon during diffusion sampling.

**Strengths:**

The paper is well-written and addresses an interesting direction by attempting to include geometric curvature information in diffusion models. The empirical results are solid and demonstrate clear quantitative and qualitative improvements, while the proposed method is computationally practical and can be plugged into existing samplers without retraining.

**Weaknesses:**

1. The paper’s technical novelty is somewhat limited. The approach primarily combines well-known techniques HVP and CG within the diffusion framework, and the underlying idea of implicit preconditioning via second-order information has been studied in related optimization and dynamics contexts. Please correct me if I am wrong about this.

2. Some statements look weird. See questions below.

**Questions:**

1. At the bottom of page 1, the paper states that “the forward diffusion transforms from Gaussian to images,” which seems reversed since the forward process typically maps data to noise, and the reverse process reconstructs images from Gaussian noise. Could you clarify if this is an error or an intentional reinterpretation of the forward/reverse process?

2. the paper attributes the severity of the zigzag phenomenon to the non-uniform curvature of the data manifold, but this explanation may not be fully accurate. Since $p_t$ ends to become smoother with increasing $t$, the zigzag behavior’s variation over timesteps cannot be entirely explained by manifold curvature. Could you provide a more rigorous justification?

---

### Official Review · Reviewer_Ceja · 2025-10-31

**Soundness:** 3
**Presentation:** 2
**Contribution:** 3
**Rating:** 4
**Confidence:** 4

**Summary:**

The paper introduces HILDA a new method for accelerating sampling on diffusion models. They provide insights from preconditioned langevyn dynamics and observe that trajectories in diffusion models have a curved nature, which makes simulation less efficient. Such problem can be corrected by including second order information into the sampling. To this purpose, they introduce a preconditioner into the backwards diffusion process. They then introduce an algorithm for approximating the preconditioner by using automatic differentiation, the conjugate gradient and Lanczos algorithm. Their algorithm demonstrates good results when compared with other diffusion samplers without a significant extra overhead in computation.

**Strengths:**

- They identify the "zigzag" phenomenon in diffusion
- They propose a preconditioner $M_t = c_t ( H_t + \lambda_t I)^{-1}$ to address the issue as well as natural algorithms to approximate $c_t, H_t, \lambda_t$
- They demonstrate benefits of their method when compared to standard baselines like DPMSolver

**Weaknesses:**

- The method adds new hyperparameters, $\kappa_*$ as well as the number of iterations for Lanczos algorithm. When compared to other methods this adds an extra layer of complexity
- In practice they leverage a frozen preconditioner that is independent of the position, but it is not clear how this is chosen, is it frozen for each $t$? If so, how does one compute it during sampling if it is independent of $x$
- There is no proof that including this preconditioning keeps the terminal distribution the same as $p_0$, as diffusion is not exactly Langevin it does require a separate proof
- All experimental results are done in a unconditional fashion, which limits the interpretability in the conditional setting. $M_t$ would require a different value for each condition. Could the authors provide some clarification and perhaps experimental validation on the conditional setting?
- The base models seem to have very weak performance not up to date with modern standards. For instance, the unconditional CIFAR-10 results have FID of around 20-30 but EDM already has achieved FID of 2.

**Questions:**

- It is unclear how despite the extra (m+k) differentiations HILDA still results in faster wall clocks than methods that don't have this overhead

---

### Official Review · Reviewer_3QDu · 2025-11-01

**Soundness:** 1
**Presentation:** 2
**Contribution:** 2
**Rating:** 2
**Confidence:** 4

**Summary:**

This paper proposes HILDA, a novel sampler for diffusion models. HILDA uses second-order information (implicit HVP) and conjugate gradient (CG) solver to derive a corrected sampling step. To ensure numerical stability, it uses some adaptive normalization schemes. The method can be applied to existing ODE solvers like DPM-Solver.

**Strengths:**

The primary contribution of this work is its novel integration of second-order, Hessian-free optimization with diffusion sampling. It also empirically found that first-order methods can produce inefficient trajectories (zigzag), and their method could alleviate this. Experiments showed overall performance gain on CelebA and CIFAR-10.

**Weaknesses:**

- The experiments are insufficient. It is an overstatement that the authors describe their experiments as "extensive". The validation is limited to small-scale, unconditional generation on CIFAR-10 and CelebA, using older model architectures. The method needs to be benchmarked more, for example, ImageNet64 or higher-resolution tasks. Experiments on larger models are also encouraged (I couldn't find the size of networks used in the paper)
- The lack of up-to-date baselines. The paper only compares HILDA to methods up to 2023 (e.g., DPM-Solver, UniPC). The current state-of-the-art in fast sampling, such as Consistency Models and MeanFlow, is missing.
- The wall-clock time comparisons in Table 1 lack standard deviations, making it hard to assess them. We usually expect the computing time to scale (super)linearly with steps, but HILDA is the fastest in 5 and 50 NFEs. Considering the overhead of HVP, it raises question.
- The visualizations in Figure 1 are not helpful; the authors claim they show a reduction in zigzag behavior, but the difference in the PC2/PC1 ratio plot (middle) appears marginal. The left plot only shows one instance (and one instance could have multiple ways of projections), which also raises question. It is also suggested that the authors make a Pareto frontier plot about FID and NFE.
- The theory is not well explored. The problem formulation is not well written. The proof needs to be evaluated carefully. For example, Lemma 2 considers a "frozen-time" SDE, while diffusion is not "frozen-time". There are many design choices that are not shown to be optimal.

**Questions:**

Many are mentioned in Weaknesses. Plus,
- For adaptive damping, when condition numbers are extremely high, the preconditioner degenerates to scaled identity and loses second-order information. How frequently does this occur, and doesn't this undermine the core motivation?
- The paper lacks failure case analysis. When does HILDA fail? How does performance degrade with poorly conditioned Hessians after damping? Ablation studies on sensitivity are encouraged.

---

### Official Review · Reviewer_Ypwz · 2025-11-01

**Soundness:** 2
**Presentation:** 2
**Contribution:** 2
**Rating:** 2
**Confidence:** 3

**Summary:**

This paper introduces HILDA, a training-free sampler for diffusion models. It aims to address the "zigzag phenomenon" in sampling trajectories. The method uses a geometric preconditioner that incorporates second-order Hessian information. This preconditioner features adaptive damping and spectral normalization. HILDA is designed as a plug-and-play module for existing ODE solvers like DPM-Solver. Experiments on CIFAR-10 and CelebA-HQ show improved FID scores over several baseline samplers.

**Strengths:**

*   The identification and analysis of the "zigzag phenomenon" is an interesting and well-motivated starting point for the work.
*   The proposed HILDA method is theoretically grounded, with a comprehensive design of the geometric preconditioner, including Hessian symmetrization, adaptive damping, and spectral normalization.

**Weaknesses:**

*   **Contradictory Claims on Computational Overhead:** There is a significant contradiction regarding the computational cost of HILDA, which severely undermines the credibility of the method. In the methodology section (Lines 342-345), it is stated that each timestep requires "6 ∼ 11 symmetrized HVPs" and that "each symmetrized HVP costs roughly twice a score evaluation." This implies a **12x to 22x** increase in computational cost per step, which is substantial and cannot be ignored. However, in the experiment section (Lines 427-428), it is claimed that the "additional computational cost incurred by our HILDA method is virtually negligible," and Table 1 shows wall-clock times similar to baselines. This contradiction must be clarified and resolved.
*   **Weak and Potentially Flawed Baselines:** The experimental baselines and reported results are not competitive with the current state-of-the-art, making it difficult to assess the method's true effectiveness.
    *   Regarding Table 2 (CelebA-HQ 256x256 with a Latent Diffusion Model), the best FID reported is over 60, which is vastly inferior to the FID of 5.15 reported in the original LDM paper [1]. The generated samples in Figure 3 also exhibit visible artifacts. While the number of function evaluations (NFE) used here (5-50) is lower than in [1] (500 NFE), the results at 50 NFE should not be this poor, raising concerns about the implementation or sampling configuration.
    *   Regarding Table 3 (CIFAR-10), the best FID is above 5, which is much worse than the FID of 1.97 achieved by EDM [2]. Comparisons with more recent and stronger baselines are lacking.

[1] Rombach, Robin, et al. High-resolution image synthesis with latent diffusion models. CVPR 2022.

[2] Karras, Tero, et al. Elucidating the design space of diffusion-based generative models. NeurIPS 2022.

**Questions:**

*   Line 91 mentions experiments on "text-guided conditional generation," but I could not find these results in the paper. If I have overlooked them, could the authors please point out their location? If they are indeed missing, could the authors provide these results, as they are important for demonstrating the generality of the method?

---

### Note · Program_Chairs · 2026-01-17
**Submission Desk Rejected by Program Chairs**

The following references in this submission do not refer to real documents and/or have major errors in bibliographic information:

 V. Fachinotti, A. Anca, and A. Cardona. A method for the solution of certain problems in least squares. International Journal for Numerical Methods in Biomedical Engineering, 27(4):595.